# Augmenting Protein Network Embeddings with Sequence Information

## Abstract

Computational methods that infer the function of proteins are key to understanding life at the molecular level. In recent years, representation learning has emerged as a powerful paradigm to discover new patterns among entities as varied as images, words, speech, molecules. In typical representation learning, there is only one source of data or one level of abstraction at which the learned representation occurs. However, proteins can be described by their primary, secondary, tertiary, and quaternary structure or even as nodes in protein-protein interaction networks. Given that protein function is an emergent property of all these levels of interactions in this work, we learn joint representations from both amino acid sequence and multilayer networks representing tissue-specific protein-protein interactions. Using these hybrid representations, we show that simple machine learning models trained using these hybrid representations outperform existing network-based methods on the task of tissue-specific protein function prediction on 13 out of 13 tissues. Furthermore, these representations outperform existing ones by 14% on average.

## 1 Introduction

Proteins can be described by their primary, secondary, tertiary, and quaternary structure or even as nodes in protein-protein interaction networks (Creighton, 1993). Some proteins with similar sequences play similar roles; others with high levels of sequence similarity can play different roles. To add further nuance, the same protein can play different roles depending on the tissue it is in and the state of that tissue. Understanding the relationship between these different levels of structure and the role that a protein plays is one of the grand challenges of biology. Recent availability of high-throughput experimental data and machine-learning based computational methods can be useful for unveiling and understanding such patterns.

We frame the problem of understanding the relationship between these complementary data sources and tissue-specific protein function as one of developing protein embeddings on top of which simple machine learning models can be trained to map a given protein to its tissue-specific function.

In this work we constructed new protein representations combining different levels of abstraction. More specifically, we constructed a 128-dimensional vector for each protein where the first 64 dimensions are derived from the amino acid sequence and the remaining 64 dimensions are obtained from embedding the protein into a tissue-specific protein-protein interaction networks. Such representations are then used to train a simple linear classifier to predict tissue-specific protein function. This approach outperforms network-based approaches which usually only use information from the protein-protein interaction network.

The main contribution of this paper include:

- Approaching the problem of tissue-specific protein function prediction from the angle of representation learning using information ranging from amino acid sequence to multilayer networks including tissue-specific protein-protein interaction
- Experimentally showing that such representations outperform network-based methods on 13 out of 13 tissues for which we perform the experiments. The best method outperforms current ones by 14% on average.

- An ablation analysis that demonstrated that our state-of-the-art results are a result of the joint embeddings

## 2 RELATED WORK

Computational methods to predict the function of proteins fall into several categories. An important step of the pipeline is developing representations for proteins. Most existing methods focus on one level of biological abstraction and develop a representation specific to this level. For example, when looking at the primary structure, the first attempt to computationally predict the role of a protein is through sequence homology. That is, using a database of protein whose sequence and function is known, methods using string similarity will find the closest proteins and use heuristics to make a prediction based on such similarity. These methods use dynamic programming and hierarchical clustering to align multiple sequence to perform homology and find the distance of a given protein to multiple proteins stored in a database. (Feng & Doolittle, 1987) (Corpet, 1988) (Corpet, 1988) (Edgar, 2004)

Beyond sequence homology, local polypeptide chains are grouped under patterns called protein domains (Bateman et al., 2004). Protein domains evolve independently of the rest of the protein chain. They are often thought of as evolutionary advantageous building blocks which are conserved across species and proteins. The presence of such building blocks in protein is used as a proxy to infer function and protein family. Pfam is a database of protein families that includes their annotations and multiple sequence alignments generated using hidden Markov models and has 17,929 families used to characterize unknown on the basis of motif presence.

Recently, inspired by the methods used in natural language processing, researchers have developed character-level language models by training algorithms such as long short-term memory (LSTM) (Hochreiter & Schmidhuber, 1997) networks to predict the next amino acid given the previous amino acids. Many recent works have gone into training and investigating the properties learned by such language models and found that they encode many biochemical properties and can be used to recover protein families. More specifically UniRep (Alley et al., 2019) uses a multiplicative LSTM (Krause et al., 2016) trained to perform next amino acid prediction on 24 million UniRef50 (Suzek et al., 2007) amino acid sequences. The trained model is used to generate a single fixed-length vector representation of the input sequence by globally averaging intermediate mLSTM numerical summaries. SeqVec (Heinzinger et al., 2019) works by training bi-directional language model ELMo (Peters et al., 2018) on UniRef50. While such models are useful descriptors and encoders of biochemical properties, they lack the local context needed to infer protein function.

While all previously-cited methods develop representations of proteins with the basic molecular components, other methods treat proteins like social networks. Proteins rarely accomplish a function in isolation and need to bind with other proteins, in a specific tissue in a given state to accomplish a function. Using this insight, many methods describe proteins using such signals. That is, using a "guilt by association principle," they take the perspective that the role of a protein can be inferred from understanding which other proteins it interacts with (Letovsky & Kasif, 2003) (Vazquez et al., 2003) (Mostafavi et al., 2008). Representation learning methods formalizing such principles usually take as input a protein-protein interaction network represented as a graph and use methods such as matrix decomposition (Tang et al., 2011) and node embeddings (Grover & Leskovec, 2016) to develop a vector representation grouping neighboring nodes into a similar position. However, these methods do not take into account the rich information that can be learned by examining a protein's primary sequence. We aim to synthesize the previous approaches, and also take more contextual information about the tissues in which proteins interact. We use OhmNet (Zitnik & Leskovec, 2017) to include the tissue hierarchy and develop tissue-specific node embeddings taking into account local neighborhoods among proteins as well as local neighborhoods among tissues.

## 3 METHODS

The main idea we present is to integrate information at different levels of the biological hierarchy into the learned representation of each protein. We used information from two sources: the amino acid sequence and the tissue-specific protein-protein interaction network. We combined these representations by concatenating them into a 128 dimensional vector and trained a linear classifier to

predict tissue-specific protein functions in a one vs all fashion. That is, each classifier is a binary classifier to predict if a given protein plays a given role in a specific tissue. We measure the area under the curve for each classifier and average it to have a tissue-specific AUROC.

### 3.1 AMINO ACID SEQUENCE REPRESENTATION

To represent the amino acid sequence, we used recent works such as UniRep and SeqVec treat the amino acids as an alphabet and the amino acid sequence as a string in that discrete alphabet. They learn representations by leveraging the millions of protein sequences available to train a machine learning model to predict the next amino acid given the previously seen amino acids. More specifically UniRep uses a multiplicative LSTM train to perform next amino acid prediction on 24 million UniRef50 amino acid sequences. The trained model is used to generate a single fixed-length vector representation of the input sequence by globally averaging intermediate mLSTM numerical summaries. SeqVec works by training bi-directional language model ELMo on UniRef50.

### 3.2 TISSUE-SPECIFIC PROTEIN NETWORK EMBEDDING

For the second source of representation, we used two different methods: Ohmnet and Node2Vec. Node2vec learns a mapping of nodes to a low-dimensional space of features that maximizes the likelihood of preserving network neighborhoods of nodes.

OhmNet encourages sharing of similar features among proteins with similar network neighborhoods and among proteins activated in similar tissues.

Given that the task of tissue-specific protein function prediction is introduced in OhmNet and uses 128 dimensional vector to compare it with other methods, all of our vectors are also constructed to produce 128 dimensional vectors.

### 3.3 DUMMY VECTORS

To perform controlled experiments that ablate various sources of information, we constructed dummy vectors that we concatenated with either the amino acid sequence representation or the tissue-specific protein network embedding. These vectors are: Random64, a 64 dimensional random vector where each dimension is generated by sampling from a uniform distribution in the [-1,1] interval. Random128 is the corresponding 128 dimensional random vector. 0-pad, which simply pads the remaining dimensions with 0s.

## 4 EXPERIMENTS

### 4.1 EXPERIMENTAL SETUP

The goal of each experiment is to solve a multi-label binary classification problem. Each label is binary and represents a specific function (more precisely a cellular function from the Gene Ontology) in a specific tissue. On each tissue, we aim to match every active protein with zero, one or more tissue-specific functions. Using a multi-output linear classifier model, we then, for each tissue, use a separate linear classifier to predict every single protein functional activation.

We evaluate and compare the protein representations from the original Ohmnet versus the augmented versions introduced in this paper. In this experiment, we run a 10-fold cross-validation with each method over 13 complex tissues (those mapped with more than one function in the Gene Ontology). Prior to that a random oversampling is run on the training data to make for the class imbalance present in almost all tissues. With each fold, the protein embeddings are split between training set (90%) and validation set (10%) in a randomly stratified fashion. This training/test split ration is done to reproduce the OhmNet setting. The task at hand is to predict the unseen validation set after fitting the training set. The name of the representations includes the data sources used to generate the 128 dimensional vectors. More details including scores for specific tissues are available in the appendix.

Out of the 13 tissues we've tried. Some highlight results include:

- Node2Vec-SeqVec outperforms Node2Vec 13/13 times

Table 1: Average AUROC for tissue-specific protein function prediction (10-fold cross validation)

|  | Ohmnet128 | Node2Vec128 | Ohmnet-SeqVec | Node2Vec-SeqVec | Ohmnet-Random | Random128 |
|---|---|---|---|---|---|---|
| AUROC | 0.52 | 0.53 | **0.6** | 0.58 | 0.51 | 0.5 |

- Node2Vec-SeqVec outperforms SeqVec 8/13 times
- Node2Vec-SeqVec achieves best performance 6/13 times
- Ohmnet-Seqvec achieves best result 2/13 times and is a very close second 3/13 times.
- Ohmnet-Seqvec scores consistently higher than both Ohmnet and Seqvec separately (respectively 11/13 times and 9/13 times)
- Either hybrid representations (Node2Vec-SeqVec and Ohmnet-Seqvec) therefore achieve best performance 8/13 times
- On average Ohmnet-SeqVec has a 14% higher AUROC than pure OhmNet
- On average Node2Vec-SeqVec has a 11% higher AUROC than pure Node2Vec

Looking at how Ohmnet-SeqVec and Node2Vec-SeqVec performs (a similar trend is observed for UniRep) shows that both Unirep and Seqvec add significant and new information that's not captured by tissue hierarchy or protein-protein interaction alone.

The average AUROC score from Random is a big higher than what could be expected from such representations thanks to the spikes (Placenta, Epidermis) which might also result from the huge functional class imbalance within those two tissues which, given the uniformity of the data, gets them more often than not on the right side of the hyperplane. Another explanation might be the low amount of data (respectiveley 35 and 72 active proteins) available on those two tissues !

## 5 CONCLUSION

In this work, we have looked at how conceptually different representations of proteins could interact and complement each other for the task of predicting function. We have shown that by merging information from two task-independent representations of proteins, we make consistently better tissue-specific function predictions in 13 complex tissues. Our ablation analysis demonstrates that the improved results are a consequence of integrating information from different levels of the biological hierarchy.

## 6 DISCUSSION/FUTURE WORK

This work explores various ways of learning representations of proteins to understand protein function in its given biological context. One key takeaway is that combining representations from different level of biological abstractions leads to improved representations as judged by their ability to predict tissue-specific protein function. Recent work on developing representation from amino acid sequence enables us to take advantage of the vast amount of unlabeled sequences and work directly with proteins whether or not they have been aligned with existing sequences or annotated using known families.

In the current experimental setting, we only focused on 13 tissues which had more than 2 functions and between 90 and 1400 active proteins. Further work can be done by looking at a more comprehensive set of tissues and functions. Additionally, we trained relatively simply classifiers in a one vs. all manners; more powerful approaches using complex models should naturally be explored.

Recent work has also developed embeddings encoding 3D protein structure. These embeddings are currently missing in this work and could also be integrated in subsequent work to help understand the relative importance of sequence, structure and protein interaction network to predict tissue-speicifc function.

We hope that our work spurs more research in representations that integrate information from multiple levels of the biological hierarchy and provide insight into the function of proteins and cells.

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

## A   APPENDICES

| tissue | Proteins | Ohmnet128 | N2vec | Ohmnet64 | N2vec64 | Ohmnet-rand | Ohmnet-seqvec | Ohmnet-unirep |
|---|---|---|---|---|---|---|---|---|
| blood | 741 | 0.55 | 0.64 | 0.56 | 0.64 | 0.49 | 0.62 | 0.62 |
| blood_vessel | 267 | 0.51 | 0.55 | 0.47 | 0.54 | 0.5 | 0.6 | 0.58 |
| bone | 90 | 0.6 | 0.53 | 0.54 | 0.52 | 0.45 | **0.67** | 0.58 |
| brain | 148 | 0.54 | 0.57 | 0.54 | 0.59 | 0.52 | 0.58 | **0.6** |
| cartilage | 45 | 0.53 | 0.48 | 0.45 | nan | 0.45 | 0.49 | 0.54 |
| central_nervous_system | 111 | 0.41 | 0.52 | 0.46 | 0.51 | 0.57 | 0.55 | **0.64** |
| epidermis | 72 | 0.53 | 0.5 | 0.52 | 0.51 | 0.49 | 0.6 | 0.55 |
| eye | 81 | 0.44 | 0.53 | 0.47 | 0.59 | 0.44 | 0.61 | 0.56 |
| kidney | 127 | 0.55 | 0.58 | 0.52 | 0.54 | 0.5 | 0.65 | 0.62 |
| leukocyte | 625 | 0.56 | 0.58 | 0.56 | 0.6 | 0.48 | **0.61** | 0.57 |
| lymphocyte | 318 | 0.58 | 0.6 | 0.59 | 0.58 | 0.49 | **0.6** | 0.57 |
| nervous_system | 611 | 0.53 | 0.52 | 0.53 | 0.54 | 0.5 | 0.6 | 0.56 |
| placenta | 35 | 0.49 | 0.34 | 0.44 | 0.39 | 0.78 | nan | 0.24 |
| **Avg. AUROC** | | 0.52 | 0.53 | 0.51 | 0.55 | 0.51 | **0.6** | 0.56 |
| **Std. AUROC** | | 0.24 | 0.23 | 0.23 | 0.23 | 0.23 | 0.24 | 0.24 |

Table 2: Average test scores per tissue for each representation after 10-fold cross-validation run

| tissue | Proteins | N2vec-seqvec | N2vec-unirep | Seqvec | Unirep | Random128 |
|---|---|---|---|---|---|---|
| blood | 741 | 0.65 | 0.62 | 0.6 | 0.61 | 0.49 |
| blood_vessel | 267 | 0.6 | 0.58 | 0.59 | 0.61 | 0.51 |
| bone | 90 | 0.58 | 0.6 | 0.65 | 0.57 | 0.51 |
| brain | 148 | 0.56 | 0.6 | 0.57 | 0.58 | 0.57 |
| cartilage | 45 | 0.43 | 0.42 | 0.46 | 0.59 | 0.55 |
| central_nervous_system | 111 | 0.55 | 0.56 | 0.56 | 0.58 | 0.45 |
| epidermis | 72 | 0.54 | 0.47 | 0.61 | 0.57 | 0.58 |
| eye | 81 | 0.64 | 0.49 | 0.58 | 0.56 | 0.41 |
| kidney | 127 | 0.65 | 0.62 | 0.68 | 0.62 | 0.48 |
| leukocyte | 625 | 0.62 | 0.59 | 0.6 | 0.58 | 0.49 |
| lymphocyte | 318 | 0.6 | 0.57 | 0.57 | 0.56 | 0.51 |
| nervous_system | 611 | 0.61 | 0.54 | 0.59 | 0.53 | 0.48 |
| placenta | 35 | 0.56 | 0.49 | nan | 0.42 | nan |
| **Avg. AUROC** | | 0.58 | 0.55 | 0.59 | 0.57 | 0.5 |
| **Std. AUROC** | | 0.23 | 0.23 | 0.24 | 0.25 | 0.23 |

Table 3: Average test scores per tissue for each representation after 10-fold cross-validation run (rest)

