# OpenReview forum: "Combining graph and sequence information to learn protein representations"
_ICLR.cc/2020/Conference — Reject_

### Official Review · AnonReviewer3 · 2019-10-22
**Official Blind Review #3**

**Rating:** 1

**Review:**

This work tries to predict the protein functional activation on a tissue by combining the information from amino acid sequence, and tissue-specific protein-protein interaction network. The authors claim that with this joint representation, their model outperforms current methods (Omhnet) on 10 out of 13 tissues by a larger margin(19% on average).

Notations:
The notations in experiment is a little bit confusing. In Table 1, the authors refer to different representations with Ohmnet128, Ohmnet64, Ohmnet-Unirep, etc. However, these are not consistent to the ones introduced in Section 4.1: Ohmnet, Ohmnet64-Unirep64, etc. And "0-pad" is introduced in section 3.3 while they denote one method as "Ohmnet64-0Padded" in section 4.1. It would be difficult for the reader to infer the meaning of these abbreviations.

Method:

--amino acid sequence representation:
It would be better to report the explained variance when using Principle Component Analysis (PCA) to project the 1024-dimensional output vector of SeqVec to 64 dimensional space.  And the authors can show us more results of different projected dimensions (with different explained variance of the PCA).

Experiments:

--model:
Maybe the authors can provide us more information about the model they use. For classification, what exactly the linear model is? For learning representation, is there any modification of the structure and hyperparameter of UniRef, SeqVec and OhmNet? And is there any regularization? Showing training details like batch size, epochs would be helpful, too.

--data:
It would be better to show the details of the data this paper uses, like what the data looks like, what is the size, the distribution, and the pre-processing. What's more, since validation set is used for tuning, it would be better to report the results on test set.

--result:
In the second paragraph of Section 4.1, it would be more clear to use a table instead of words to show the results. What's more, what's exactly the 13 tissues this paper is using? Why they are chosen? Exactly what is the AUROC of each protein in each tissue?  What the learning curves look like?

Another big issue is, what "current methods" is this paper comparing its result with? It seems like the authors are comparing their implementation of Ohmnet-SeqVec + linear model with Ohmnet + linear model, and report that the former one is of 19% higher AUROC than the latter. But how about the results of other models/methods on the same task in the literature. Is there anyone using similar joint representation and what is their results?

--conclusion:
Since the proposed methods only achieve best results  in 10 out of 13 tissues, it is improper to claim "… we make consistently better tissue-specific function predictions in 13 complex tissues …".

In conclusion, I find this is an interesting paper, that the authors tries to combine amino acid sequence representation and tissue information to predict the activation of protein on specific tissue. However, the authors should perform more rigorous experiment, and show us more implementation details. What's more, comparing results with the start-of-art methods on the same task setting is important, too.



**Experience Assessment:**

I do not know much about this area.

**Review Assessment: Checking Correctness Of Derivations And Theory:**

N/A

**Review Assessment: Checking Correctness Of Experiments:**

I carefully checked the experiments.

**Review Assessment: Thoroughness In Paper Reading:**

I read the paper at least twice and used my best judgement in assessing the paper.

---

### Official Review · AnonReviewer2 · 2019-10-23
**Official Blind Review #2**

**Rating:** 1

**Review:**

This paper introduces a method to incorporate both sequence information and graph information to learn the protein representations. The idea is very straightforward. Basically, it used the embedding from OhmNet [Marinka et al, 2017] for the graph information and used the sequence information from UniRep [Ethan et al, 2019] or SeqVec [Michael et al, 2019]. It uses one experiment to show the performance of the combination of the two pieces of information.

This paper should be rejected for the following reasons:
(1) The paper is obviously in the preliminary form without too much polish.
 (2) The simple combination of the results from two published articles is not that interesting
(3) the presentation of the paper and idea is not in an acceptable form (the authors should at least draw a figure to show the big idea of the paper).
(4) the experiment is not convincing (there is only one experiment and it is not compared with the other state-of-the-art methods; since an embedding of a protein can be of broad usage, the authors should give its performance on four tasks: protein function prediction (GO term) [Maxat et al, 2018], enzyme function prediction (EC number) [Yu et al, 2018], protein secondary structure prediction [Sheng et al, 2016], protein contact map prediction [Jinbo Xu, 2019])
(5) The learned embedding is not well discussed. The author should at least visualize the embeddings and check the physical and biological meaning of those embeddings, if possible.

Since this manuscript would be for sure and have to be largely rewritten in the future, I would not give too many detailed suggestions but some high-level suggestions if the authors would like to refine this manuscript further and submit it somewhere else or ICLR next year:
(1) Further improve the idea of combining different sources of information. Combining different pieces of information will definitely be helpful but the authors should figure out a way to use them in a more natural way.
(2) Compared with other methods, which can combine different sources of information.
(3) Run more experiments on various tasks instead of one: protein function prediction (GO term), enzyme function prediction (EC number), protein secondary structure prediction, protein contact map prediction
(4) Refine the representation of the paper.

References:
[Marinka et al, 2017] Predicting multicellular function through multi-layer tissue networks, 2017, https://arxiv.org/abs/1707.04638
[Ethan et al, 2019] Unified rational protein engineering with sequence-based deep representation learning, 2019, Nature Methods
[Michael et al, 2019] Modeling the Language of Life – Deep Learning Protein Sequences, 2019, https://www.biorxiv.org/content/10.1101/614313v2
[Maxat et al, 2018] DeepGO: predicting protein functions from sequence and interactions using a deep ontology-aware classifier, 2018, Bioinformatics
[Yu et al, 2018] DEEPre: sequence-based enzyme EC number prediction by deep learning, 2018, Bioinformatics
[Sheng et al, 2016] Protein Secondary Structure Prediction Using Deep Convolutional Neural Fields, 2016, Scientific Reports
[Jinbo Xu, 2019] Distance-based protein folding powered by deep learning, 2019, PNAS

**Experience Assessment:**

I have published one or two papers in this area.

**Review Assessment: Checking Correctness Of Derivations And Theory:**

I carefully checked the derivations and theory.

**Review Assessment: Checking Correctness Of Experiments:**

I carefully checked the experiments.

**Review Assessment: Thoroughness In Paper Reading:**

I read the paper thoroughly.

---

### Official Review · AnonReviewer1 · 2019-10-24
**Official Blind Review #1**

**Rating:** 3

**Review:**

In this study, the authors develop a method to predict the function of proteins from their structure as well as the network of proteins with which they interact in a given tissue. The method consists in training a linear classifier on the output of two existing embedding methods, UniRep/SeqVec and OhmNet, respectively embedding the amino acid sequences and the tissue-specific protein-protein interaction networks. This method improves prediction of protein function by 19% compared to OhmNet alone.

Although the topic is important and the article clearly written, I would tend to reject this article because there is no innovation in ML that would justify presentation at ICLR.

Strengths:
- the article is well-written and straight-forward. Prior art is well-described.
- timely and important topic (prediction of protein function), where ML is likely to have an big impact.
- positive scientific result (prediction is improved compared to prior art).

Weakness:
- the ML aspect of this work is entirely based on prior art, the main innovation consisting in fitting a linear classifier on concatenated features extracted by two existing embedding methods (UniRep/SeqVec and OhmNet).


Additional feedback:
- In the ablation studies, why not include the condition SeqVec-Random and UniRep-random?
-"The average AUROC score from Random is a big higher than what could be expected from such representations thanks to the spikes (Placenta, Epidermis) which might also result from the huge functional class imbalance within those two tissues which, given the uniformity of the data, gets them more often than not on the right side of the hyperplane. "
=> unclear sentence.
- "is a big higher" => typo
- "beta sheets ." => typo



**Experience Assessment:**

I do not know much about this area.

**Review Assessment: Checking Correctness Of Derivations And Theory:**

N/A

**Review Assessment: Checking Correctness Of Experiments:**

I assessed the sensibility of the experiments.

**Review Assessment: Thoroughness In Paper Reading:**

I read the paper thoroughly.

---

### Public Comment · ~Christian_Dallago1 · 2019-10-11
**Interesting idea but needs major rework**

The approach described in this work fuses protein information (obtained via embeddings) and network information (obtained via observed protein-protein interactions) to predict protein function according to GO. The proposed method betters the existing ones by a margin of 19%.

- About the writing:

In general, I find the manuscript is written in good English. Nevertheless, at times, I feel more consideration should have been put in the choice of words in order to ease a non biological audience into the manuscript. En example of this is "Proteins are generally understood through four levels of structures", and following use of the word "structure" (e.g. in primary structure) in the Introduction. In protein space, structure has a very precise meaning. I would have personally preferred the word representation, or alike. On this note: sometimes the writing gets rapidly technical, and I fear that the audience might not fully grasp important concepts. An example of this is the description of quaternary structure in the introduction, which could be easily explained by simply describing proteins forming interactions amongst themselves or other proteins. Furthermore, the authors seem to sometimes loose consistency, e.g. UniRep becomes Unirep, SeqVec becomes Seqvec, OhmNet becomes Ohmnet. There also seem to be artifacts of editing, e.g.:

... thanks to spikes (Placenta, Epidermis) ....

but never before have actual tissues been discussed and there is no reference to a figure which might display said spikes. More on this to follow.

Another example are the two last paragraphs of the Introduction, which share the same beginning "In this work,..", where instead I would have expected a natural following of the last paragraph to the second last.

- Background/related work:

While this particular approach (embeddings+graph) is the first I've heard of, I can hardly imagine there not existing any other approach using a mix of protein-level (1D) and network-level (4D) information. It would have been nice to see at least one paragraph spent on any previous work that aims at solving protein function prediction using these information.

I find the manuscript is generally lacking in citations, breath and detail. For example, the authors describe sequence homology (as a way to infer function), but never reference important work introducing or exploring this concept (from the top of my head: Sander 1991, Rost 1992). Another example, in the Introduction, relates to the concept of secondary structure, which is described in Kabsch&Sander 1983.

Some claims (e.g. in the Introduction) lack proper validation (either external or in the manuscript), e.g. "Recent availability of high-throughoutput experimental data and machine learning based computational methods can be useful for unveiling and understanding such patterns.": Why? Where is this shown?

- Science & results:

While I see the appeal of the approach, I am not convinced that the authors are currently able to prove this. I am lacking many things: a proper description of the goal (I personally get confused about tissue-specific functions vs. GO, vs. prediction of the tissue,...), hard numbers on the targets (binary classification of GO terms: how many? which ones? Supplementary table),...

In my general confusion, I stumbled across a giant red flag for protein-related prediction tasks: the split of test and training sets. The authors describe this as a stratified random split. I sincerely hope that the stratification has NOT been made by looking at homology, and instead I wish the authors had discussed reduction of homology in their training and test sets, potentially picking very far related proteins between the different sets. Not to mention: I'm missing the size of test and train, and what "stratification" means in this context, how many samples per label,...

Dummy vectors: I'm not sure that the boundaries in the vector spaces of OhmNet and SecVeq are [-1, 1].

The authors at some point introduce a "Ohmnet64-0Padded" and "Ohmnet64-Random64" feature vectors, but I was unable to find any results for using these in the manuscripts.

In Table 1: I'm missing something that approximates an error estimate. I would have preferred to see the actual curves, in order to gaze an understanding about how their behavior in different conditions of sensitivity/specificity.

As mentioned above, "spikes" are mentioned, but no graph is presented. I'm missing the exact tissues for which each predictor was better (e.g. 6/13 and 4/13 --> hard numbers don't tell me if they are not overlapping).

These last considerations make me personally very suspicious about the results presented.

- Conclusion

I find the manuscript explores an interesting idea, but I *urge* the authors to rework the manuscript from top to bottom to (i) explain the background better, (ii) explain the problem they are trying to solve better, (iii) present the datasets, labels, classes better, and (iv) the results more clearly.

---

### Decision · Program_Chairs · 2019-12-19

**Decision:**

Reject

**Comment:**

The paper presents a linear classifier based on a concatenation of two types of features for protein function prediction. The two features are constructed using methods from previous papers, based on peptide sequence and protein-protein interactions.

All the reviewers agree that the problem is an important one, but the paper as it is presented does not provide any methodological advance, and weak empirical evidence of better protein function prediction. Therefore the paper would require a major revision before being suitable for ICLR.